# Resistance Assessment to PHYVV and PepGMV in Wild and Domesticated Accessions of *Capsicum annuum* L. by Bioballistic Inoculation

**DOI:** 10.3390/plants14172708

**Published:** 2025-08-31

**Authors:** Karla Vanessa De Lira-Ramos, Ernesto González-Gaona, José Francisco Morales-Domínguez, Diana Lilia Trejo-Saavedra, Joaquín Sosa-Ramírez, Rafael F. Rivera-Bustamante, José de Jesús Luna-Ruíz

**Affiliations:** 1Centro de Ciencias Básicas, Universidad Autónoma de Aguascalientes, Av. Universidad 940, Aguascalientes 20100, Aguascalientes, Mexico; kvlr1820@gmail.com; 2Campo Experimental Pabellón, Centro de Investigación Regional Norte Centro del Instituto Nacional de Investigaciones Forestales, Agrícolas y Pecuarias, Kilómetro 32.5 Carretera Aguascalientes–Zacatecas, Pabellón de Arteaga 20660, Aguascalientes, Mexico; gonzalez.ernesto@inifap.gob.mx; 3Departamento de Química, Centro de Ciencias Básicas, Universidad Autónoma de Aguascalientes, Av. Universidad 940, Aguascalientes 20100, Aguascalientes, Mexico; francisco.morales@edu.uaa.mx; 4Departamento de Ingeniería Genética, Centro de Investigación y de Estudios Avanzados del IPN (CINVESTAV), Unidad Irapuato, km. 9.6 Libramiento Norte, Irapuato 36500, Guanajuato, Mexico; dtrejo@ira.cinvestav.mx (D.L.T.-S.); rrivera@ira.cinvestav.mx (R.F.R.-B.); 5Departamento de Ciencias Agronómicas, Centro de Ciencias Agropecuarias, Universidad Autónoma de Aguascalientes, Av. Universidad 940, Aguascalientes 20100, Aguascalientes, Mexico; joaquin.sosa@edu.uaa.mx

**Keywords:** geminivirus, begomovirus, bioballistics, viral resistance, symptoms, PCR

## Abstract

Chili pepper (*Capsicum annuum* L.) is an economically important crop in Mexico, with a production that is limited by viral diseases caused by Begomovirus infections such as PHYVV and PepGMV, both transmitted by *Bemisia tabaci.* These viruses affect both domesticated cultivars and wild populations. The use of resistant genotypes is the most effective strategy to reduce Begomovirus incidence. Since no commercial cultivars with resistance are currently available, in this study, 15 *Capsicum annuum* accessions with different levels of domestication were inoculated separately with infectious PepGMV and PHYVV dimers by bioballistics, in order to identify sources of genetic resistance or tolerance to these viruses. Symptom progression (severity), incidence, the area under the disease progress curve (AUDPC), and molecular detection of viral DNA by PCR were recorded in asymptomatic plants. PCR results with oligonucleotides targeting PepGMV and PHYVV showed that 96% of asymptomatic plants were positive, confirming that viral replication occurred without the development of visible symptoms. Significant differences were observed among accessions, with wild and semidomesticated accessions showing very low values of severity, incidence, and AUDPC. Notably, the wild accession Acc-106 exhibited resistance to PepGMV and tolerance to PHYVV, with minimum values of severity (0 and 0.13) and incidence (0% and 13%) at 35 days postinoculation.

## 1. Introduction

The cultivation of chili pepper (*Capsicum annuum* L.) is fundamental to both the economy and food systems in Mexico. However, its production is constrained by viral infections such as the *Pepper huasteco yellow vein virus* (PHYVV) and the *Pepper golden mosaic virus* (PepGMV), both belonging to the genus Begomovirus of the family Geminiviridae [1]. Plants infected with PHYVV typically exhibit symptoms including vein yellowing, yellow mosaic, leaf distortion, leaf curling, stunted growth, and yield reduction [2]. In the case of PepGMV, the symptoms include golden mosaic, leaf deformation, and reduction in plant size [3]. Its primary vector for both viruses is the whitefly (*Bemisia tabaci* Genn.) (Hemiptera: Aleyrodidae), which has a high transmission capacity and has developed resistance to insecticides, complicating efficient control [4,5]

In Mexico, PHYVV and PepGMV are widely distributed across *C. annuum* production areas. Both viruses have been detected in domesticated cultivars and wild populations. Their presence has been reported in northern states such as Tamaulipas and Sinaloa, as well as in central and southern regions, including Puebla and the State of Mexico [6,7,8,9]. Specifically, PepGMV has been identified in regions like Baja California, Sonora, the Comarca Lagunera, San Luis Potosí, Michoacán, Campeche, Yucatán, Puebla, and the State of Mexico [7,8,9,10,11,12].

The use of resistant genotypes is the most effective strategy to reduce *Begomovirus* incidence and minimize agrochemical use [13]. Wild relatives of cultivated species are recognized as valuable sources of resistance genes for pests and diseases [14]. Regarding wild plants, Hernández-Verdugo et al. [15] and Godínez-Hernández et al. [16] identified wild accessions of *C. annuum* and *C. chinense* with resistance to PHYVV. These species are important because they represent the genetic basis from which current commercial cultivars have been derived. For instance, transcriptomic analysis under biotic and abiotic stress in *C. chinense* led to the identification of the *CchGLP* gene in the BG-3821 accession, which encodes a germin-like protein associated with PHYVV resistance [17,18]. Furthermore, García-Neria and Rivera-Bustamante [19] demonstrated that BG-3821 displays natural resistance to both PepGMV and PHYVV, involving defense mechanisms such as systemic acquired resistance (SAR) and restricted viral replication.

Additionally, a phenomenon of symptom remission or recovery has been reported in pepper plants experimentally infected with PepGMV dimers. This recovery is associated with a specific resistance mechanism involving the accumulation of small interfering RNAs (siRNAs), suggesting the participation of post-transcriptional gene silencing (PTGS) as an antiviral defense pathway. Unlike general resistance, this response was specific to PepGMV and not observed with PHYVV, supporting the hypothesis of highly selective defense mechanisms in certain pepper accessions [20]. Despite these advances, no commercial pepper cultivars in Mexico have been confirmed to possess resistance to both PHYVV and PepGMV.

Mechanical inoculation of begomoviruses is generally inefficient or not feasible, as their transmission requires vector-mediated introduction into the phloem, and sap inoculation rarely produces consistent infections in host plants [21,22,23]. Traditionally, resistance screening has been performed through inoculation with *B. tabaci* which delivers viral particles directly into the plant’s vascular system. While effective, this method requires high technical expertise, specialized infrastructure, and extended evaluation periods [24,25]. As an alternative, bioballistic (gene gun) inoculation has proven effective for introducing viral DNA directly into plant tissues, particularly in *Capsicum* species [20,24]. This approach circumvents the limitations associated with vector- or *Agrobacterium tumefaciens*-mediated transmission, making it particularly useful for recalcitrant species [25,26].

In this study, it was proposed that pepper accessions with varying degrees of domestication would display differential resistance to PepGMV and PHYVV infection. Therefore, 15 *Capsicum annuum* accessions with different levels of domestication were inoculated separately with infectious PepGMV and PHYVV dimers by bioballistics, in order to identify sources of genetic resistance or tolerance to these viruses. The results revealed significant differences in symptom severity and disease progression among accessions and domestication levels, with wild and semidomesticated accessions generally exhibiting lower severity and Area Under the Disease Progress Curve (AUDPC), particularly in response to PHYVV. Additionally, viral DNA was detected in asymptomatic plants by PCR, indicating that some plants were able to restrict the replication or expression of these viruses. For instance, the wild accession Acc-106 showed resistance to PepGMV and tolerance to PHYVV. These findings contribute to the understanding of genetic resistance to *Begomovirus* in *C. annuum*.

## 2. Results

### 2.1. Evolution of the Symptoms of PepGMV and PHYVV

The control plants from the 15 accessions that were inoculated with tungsten particles lacking viral DNA did not exhibit any symptoms of viral infection. In contrast, plants inoculated with PepGMV and PHYVV developed characteristic symptoms, the progression of which was monitored at 7, 14, 21, 28, and 35 days postinoculation (dpi) to establish a specific severity scale for each virus. Symptom evolution was documented photographically in representative plants that displayed typical symptoms at varying degrees of severity for PepGMV (Figure 1) and PHYVV (Figure 2), and classified according to the severity scale shown described in the Section 4 Tables 4–6. Although both viruses generally induce similar symptoms in pepper plants, the descriptions accounted for the phenotypic particularities associated with each infection. The first symptoms induced by both viruses were observed starting at 7 dpi.

#### 2.1.1. Development of PepGMV Symptoms

PepGMV-resistant plants remained asymptomatic (severity = 0) throughout the 35 dpi duration of the experiment (Figure 1a). Plants that developed initial symptoms of PepGMV exhibited small bumps or yellow spots at the base of the inoculated leaves (Figure 1b). These bumps and spots expanded and became more pronounced, forming yellow mosaics on the leaf surface. The symptoms intensified and spread across the leaf, covering a larger area (Figure 1c). Subsequently, infected leaves displayed deformations and chlorosis, with yellow mosaics distributed over the entire leaf surface. The leaves became curled and showed an irregular texture (Figure 1d). In severe cases, the leaves were completely deformed, becoming elongated and wavy, with prominent yellow blotches (Figure 1e).

#### 2.1.2. Development of PHYVV Symptoms

As with PepGMV, PHYVV-resistant plants remained asymptomatic throughout the 35 dpi evaluation period (Figure 2a). Initial symptoms appeared as slight yellowing of the veins at the base of the leaves (Figure 2b). In more advanced stages, the yellowed veins formed a reticulated pattern extending from the base to the middle of the leaf. These symptoms intensified and spread across the leaf, covering a larger area (Figure 2c). In the most advanced stages, infected leaves exhibited severe deformation and chlorosis, with an intense yellow hue and a prominent network of veins affecting approximately 90% of the surface. The leaves showed a downward bend at the midsection and developed an irregular texture (Figure 2d). In extreme cases, the leaves adopted a spoon-like shape, with yellow venation distributed across the entire surface. These symptoms represented the most severe expression of PHYVV infection (Figure 2e).

### 2.2. Response of Pepper Accessions to PepGMV and PHYVV Inoculation

The results of the analysis of variance (ANOVA) for each virus allowed us to evaluate the effect of accession, evaluation time (dpi), and their interaction on the severity of symptoms induced by PepGMV and PHYVV. For both viruses, highly significant differences were observed among accessions and across evaluation times (*p* < 0.001), indicating that both factors had a substantial influence on symptom expression. The lack of significant interaction between accession and time indicates that, although accessions differed in their overall symptom severity, the temporal pattern of symptom development was consistent across all accessions. This suggests that the biological processes underlying symptom progression over time are conserved among the diverse genetic backgrounds studied. Furthermore, no accession exhibited atypical or delayed symptom development, supporting the reliability of the inoculation and scoring methods used.

Notably, the effect of accession was more pronounced in response to PHYVV (*f* = 22.495) than to PepGMV (*f* = 9.493), indicating greater differentiation in accession response to PHYVV. Likewise, the progressive increase in symptom severity over time was more marked in PHYVV-infected plants (*f* = 40.16), suggesting a more aggressive or persistent infection compared to PepGMV (*f* = 13.553).

### 2.3. Incidence and Severity of PepGMV and PHYVV Infection

Symptom assessment at 35 dpi revealed substantial variability in both incidence and severity among the 15 *Capsicum annuum* accessions inoculated with PepGMV and PHYVV (Table 1). Overall, higher incidence and severity were observed in response to PHYVV compared to PepGMV.

For PepGMV, incidence ranged from 0% (Acc-106) to 67% (Acc-12 and Acc-113), with an overall average of 40% (Table 2). The mean severity across all accessions was 0.65. The accession with the lowest severity was Acc-106 (0.0, Pico Paloma), while the highest was Acc-113 (1.33, Bola). Although the mean severity recorded for all 15 accessions in response to PepGMV did not exceed 1.4 on the 0–4 scale, the differences among some accessions were statistically significant.

In the case of PHYVV, incidence was generally higher, with a mean of 77%, ranging from 13% (Acc-106) to 100% (Acc-113). The overall mean severity was 1.10, with the lowest value again in Acc-106 (0.13, Pico Paloma), while the highest severity was observed in Acc-06 (2.30, Guajillo).

### 2.4. Influence of the Level of Domestication of Accessions on the Response to Viral Inoculation

The results of the factorial ANOVA by domestication group revealed a significant effect on symptom severity (PepGMV: *f* = 13.055, *p* < 0.001; PHYVV: *f* = 63.663, *p* < 0.001), indicating that the degree of domestication had a considerable influence on symptom expression. Specifically, wild and semidomesticated accessions exhibited lower severity levels compared to domesticated ones, suggesting greater tolerance or natural resistance in less domesticated groups. Time also had a significant effect on both virus treatments (PepGMV: *f* = 12.535, *p* < 0.001; PHYVV: *f* = 34.779, *p* < 0.001), confirming that symptom severity increased over time (dpi). In contrast, the interaction between domestication level and time was not significant (PepGMV: *f* = 0.226, *p* = 0.986; PHYVV: *f* = 1.051, *p* = 0.396), indicating that the temporal progression of symptoms was consistent within each domestication group. In other words, although symptom severity differed by domestication level, the pattern of increase or stability over time remained similar across groups.

Table 1 shows the mean severity, severity range (minimum and maximum values among accessions within each group), and percentage of incidence for domestication groups (wild, semidomesticated, and domesticated) of *C. annuum* accessions. Mean values suggested that domesticated accessions tended to exhibit higher severity and incidence against both viruses, particularly PHYVV. However, according to Tukey’s test (*p* ≤ 0.05), these differences were not statistically significant for PepGMV, and only domesticated vs. wild groups differed significantly for PHYVV. The lack of statistical separation in some cases may be due to the high variability among accessions within groups and the unequal sample sizes, which could mask differences in central tendency.

These results suggest that domestication may be associated with increased susceptibility to both viruses, with a more pronounced effect observed in the case of PHYVV. Nonetheless, the observed pattern was not uniformly supported by statistical significance, highlighting the influence of within-group variability and sample size. The difference in symptom severity between domesticated and wild accessions may be attributed to the domestication syndrome, which encompasses drastic or gradual changes in plant physiology, anatomy, and defense-related traits. Additionally, resistance or susceptibility mechanisms may vary depending on the virus species and environmental conditions.

### 2.5. Area Under the Disease Progression Curve (AUDPC)

The results of the AUDPC, calculated from the mean severity and incidence for each of the 15 accessions inoculated with PepGMV and PHYVV during the 35 dpi period, as well as the classification of the accessions, are presented in Table 3. Based on these AUDPC values, the accessions were classified according into categories of resistant (R), tolerant (T), moderately susceptible (MS), susceptible (S), and highly susceptible (HS), as described in the Section 4.

The nonparametric Kruskal–Wallis analysis for AUDPC revealed significant differences among accessions for both viruses: PepGMV (*χ*^2^ = 54.12, *df* = 14, *p* < 0.001) and PHYVV (*χ*^2^ = 43.17, *df* = 14, *p* < 0.001).

Using these categories, phenotyping enabled the establishment of specific symptom progression profiles for each accession, facilitating their classification.

For PepGMV, most accessions were classified as tolerant, three as moderately susceptible (Acc-104, Acc-105, Acc-113), and one as resistant (Acc-106). In the case of PHYVV, the response was more heterogeneous: seven accessions were tolerant, seven moderately susceptible, and one susceptible (Acc-06).

Overall, accessions inoculated with PepGMV exhibited lower AUDPC values compared to those inoculated with PHYVV, suggesting that PHYVV induced greater severity in most of the evaluated accessions.

The disease induced by PepGMV showed wide variability in AUDPC progression. Accession Acc-106 (wild, Tabasco) was the only one classified as resistant (AUDPC = 0), exhibiting no symptoms (Figure 3a Among the domesticated accessions, Acc-60 (Huacle, Oaxaca) and Acc-65 (Cascabel, Jalisco) displayed low AUDPC values (1.17 and 1.65, respectively), indicating a tolerant response (Figure 3b,c). In contrast, Acc-113 (Bola), an improved variety, recorded the highest AUDPC value (7.00), reflecting greater susceptibility (Figure 3d).

For PHYVV, increased disease progression and accumulation were observed in most accessions. Although Acc-106 (wild, Tabasco) maintained a low AUDPC value (0.93), other wild accessions such as Acc-105 and Acc-104 (both Amashito types) also showed low AUDPC values (2.33 and 5.50, respectively), being classified as tolerant and moderately susceptible (Figure 4a–c). In contrast, the domesticated accession Acc-06 (Guajillo) exhibited the highest AUDPC value (16.00) and was therefore classified as susceptible (Figure 4d).

### 2.6. Identification of PepGMV and PHYVV in Asymptomatic Plants by PCR

PCR analysis confirmed the quality of all DNA samples through successful amplification of the actin gene, ensuring reliable PCR performance. Using this validated method, PCR analysis for PepGMV and PHYVV detection revealed an infection rate of over 90% in asymptomatic plants, confirming the high reliability of the bioballistic inoculation method. Out of the 167 asymptomatic plants, 154 were analyzed, yielding a 96% rate of positive detection. Specifically, 98% of the samples tested positive for PepGMV, while 91% were positive for PHYVV. PCR analysis targeting fragments of the replication (Rep) and capsid (CP) genes for both viruses confirmed the presence of viral DNA in some asymptomatic individuals. Figure 5 illustrates the amplification of PepGMV genes in Accession Acc-106. In plants 2, 3, and 5, no amplification of the Rep gene was detected (Figure 5a), while in plant 7, the CP gene was not amplified (Figure 5b). Both genes play essential roles in the viral infection cycle Rep in replication and CP in “encapsidation”, and their lack of amplification may be associated with the activation of the plant’s defense mechanisms that interfere with viral gene expression.

The presence of viral DNA in asymptomatic plants suggests that some accessions may limit viral replication or movement, effectively suppressing symptom development despite infection. This observation aligns with resistance mechanisms reported in Capsicum species, where the plant can tolerate or restrict virus proliferation without displaying visible symptoms. Further studies quantifying viral loads and characterizing host defense responses would be valuable to elucidate these tolerance mechanisms.

## 3. Discussion

Several studies have highlighted the importance of wild *Capsicum* populations as valuable genetic sources for pathogen resistance due to their high genetic diversity and adaptability [15,27,28]. These populations, not subjected to artificial selection processes, exhibit a wide phenotypic and genotypic variability, including high levels of disease resistance [28]. This genetic diversity present in these groups has been used to identify individuals with different degrees of resistance or tolerance, which can be useful in breeding programs aimed at managing viral diseases such as those caused by PepGMV and PHYVV [29]. In this study, the use of bioballistic inoculation of infectious clones allowed a precise assessment of resistance across 15 wild, semidomesticated, and domesticated *Capsicum annuum* accessions. Our findings revealed that wild and semidomesticated accessions consistently exhibited lower levels of symptom severity and incidence compared to domesticated ones. The wild accession Acc-106 did not develop any symptoms after PepGMV inoculation and showed a tolerant response to PHYVV, with a mean severity of 0.13 and an incidence of 13% at 35 dpi. In contrast, some domesticated accessions, such as Acc-06 and Acc-113, developed more severe symptoms, reflecting higher susceptibility to viral infection. Although severity and incidence tended to increase with the degree of domestication, statistical analyses indicated that these differences were not always significant across all groups, particularly in the case of PepGMV. This lack of consistent statistical significance may be attributed to the high phenotypic variability within domestication groups and the unequal sample sizes, both of which can reduce the statistical power to detect biologically relevant differences in resistance or susceptibility. Such variability is common in *Capsicum* germplasm and underscores the importance of evaluating a broader range of accessions to more robustly detect resistance patterns associated with domestication.

It is important to highlight that alternative inoculation methods, such as whitefly (*Bemisia tabaci*)-mediated transmission, require specialized infrastructure and expertise, limiting their accessibility and consistency for large-scale screening. Additionally, Agrobacterium tumefaciens-mediated inoculation, although commonly used in other plant virus systems, is often inefficient in *Capsicum* species due to their recalcitrant nature. Preliminary experiments conducted in this study confirmed that *Agrobacterium*-mediated inoculation failed to infect any of the tested accessions, including Acc-106. Consequently, bioballistic inoculation was employed as a reliable and effective alternative, enabling direct delivery of viral DNA into plant tissues and achieving infection rates exceeding 90%. This method has been successfully applied in prior studies [10,19,20,24,25,30], supporting its suitability for resistance screening in *Capsicum* germplasm. These findings align with those of Godínez-Hernández et al. [16], who evaluated 49 populations of wild habanero pepper (*Capsicum chinense*) from Yucatán, Mexico, and reported significant resistance to PHYVV, attributed to low viral DNA accumulation and absence of visible symptoms in experimentally infected plants. Subsequently, Barrera-Pacheco et al. [17] identified the *CchGLP* gene in the wild accession BG-3821 of *C. chinense*, associating it with resistance to PHYVV. More recently, Retes-Manjarrez et al. [31,32] confirmed these findings by identifying wild populations of *C. annuum* with a high level of resistance to PHYVV, in which the presence of the *CchGLP* gene was also detected in the three lines evaluated.

Among the key mechanisms potentially underlying viral resistance is the restriction of viral movement within the plant [19,23,33]. Although viral DNA quantification was not performed, the viral Rep and CP genes of PepGMV were detected in 98% of asymptomatic plants, and those of PHYVV in 91% of asymptomatic plants, suggesting that the virus was able to replicate but not to trigger visible symptoms. This detection highlights the complexity of resistance mechanisms in *Capsicum* accessions, where plants support viral presence without developing symptoms, indicating a form of tolerance or restricted viral movement and replication.

This phenomenon has been reported previously in Begomovirus-resistant *Capsicum*, where defense mechanisms such as transcriptional gene silencing, DNA methylation, or inhibition of viral movement proteins contribute to suppressing symptom expressions despite infection. Understanding these processes is crucial for breeding strategies aiming not only at symptom reduction but also at limiting viral spread.

Furthermore, the observed difference in disease aggressiveness between PHYVV and PepGMV, reflected in AUDPC values and symptom severity, suggests that resistance or tolerance may be virus-specific, requiring evaluation of host responses to multiple viral species. This underscores the importance of integrating both phenotypic assessments and molecular diagnostics into the characterization of resistance to begomoviruses in pepper germplasm. This observation aligns with previous studies indicating that begomovirus resistance may involve limited viral replication or activation of host defense responses that suppress viral DNA accumulation in infected tissues [4,16]. These resistance mechanisms have been further associated with transcriptional and post-transcriptional gene silencing, DNA methylation, and inhibition of viral movement proteins within the host plant [10,19,23,30]. Although absolute viral DNA levels were not quantified in our study, the presence of the virus in asymptomatic plants suggests that certain accessions, particularly wild ones, may possess mechanisms that limit viral replication or spread to undetectable levels, which coincides with what has been reported in resistant accessions of *Capsicum* in previous studies [15,16,19,34].

Our results show that symptom severity and incidence induced by PepGMV and PHYVV were significantly influenced by domestication level when evaluated through the Area Under the Disease Progression Curve (AUDPC). Statistical analyses of individual accessions revealed significant differences in AUDPC values, with wild accessions generally exhibiting lower symptom progression and incidence compared to many domesticated accessions. However, when grouped by degree of domestication, differences among the domesticated (D), semidomesticated (SD), and wild (W) groups were not always consistently significant, likely due to considerable phenotypic variability within groups and unequal sample sizes. This trend is consistent with what was reported by Hernández-Verdugo et al. [15], who found high levels of begomovirus resistance in wild *Capsicum annuum* accessions, and with what was pointed out by Meyer and Purugganan [28], who argue that the domestication process may reduce the genetic variability associated with defense genes.

Regarding symptom development, both viruses (PepGMV and PHYVV) induced visible symptoms from 7 dpi, with more pronounced and persistent symptoms observed in PHYVV-infected plants. Systematic photographic documentation, along with the development of a pictorial scale of severity, allowed for a detailed description of symptom evolution and the establishment of reliable visual categories. This five-level scale (ranging from 0 to 4) not only complements previously published scales by Anaya-López [34] for PepGMV and by Torres-Pacheco [35] for PHYVV but also provides a visual framework adapted for evaluating *Capsicum* under controlled conditions.

The wide phenotypic variability observed among accessions in response to both viruses reflects the underlying genetic diversity and differing domestication levels of the evaluated materials. In this context, visual phenotyping, complemented with the calculation of the area under the disease progression curve (AUDPC), offered a robust tool to integrate the temporal dynamics of the disease progression. This phenotyping strategy, widely validated in plant pathology [36,37], enabled the detection of differentiated patterns in the progression of symptom severity and incidence. Accessions inoculated with PHYVV tended to develop higher AUDPC values than those inoculated with PepGMV, confirming a more aggressive infection by the former. Once again, accession Acc-106 stood out as the only one with zero or minimal AUDPC values, reinforcing its classification as resistant. Together, systematic visual characterization using a pictorial scale, combined with quantitative analysis through AUDPC, strengthens the selection process of resistant or tolerant genotypes and provides a solid methodological foundation for future breeding efforts and resistance evaluations against begomoviruses.

## 4. Materials and Methods

### 4.1. Plant Material

The pepper accessions evaluated correspond to genetic material stored in the Germplasm Bank of the Centro de Ciencias Agropecuarias de la Universidad Autónoma de Aguascalientes. We selected 15 accessions of *Capsicum annuum* with high genetic diversity, originating from different regions of Mexico and exhibiting various degrees of domestication: wild, semidomesticated, domesticated, or cultivated. This set includes *Criollo de Morelos 334* (CM334), an accession known for its resistance to *Phytophthora capsici* (Table 4).

### 4.2. Plant Production

The seeds of each accession were germinated following the procedure proposed by García-Nevárez et al. [38] with some modifications. A pregermination treatment with 10% NaOH was applied for 4 min followed by four 1 min rinses with distilled water. The seeds were then soaked in a solution of 5000 mg/L of gibberellic acid (Bio Gib^®^) for 24 h at 25 °C. Afterward, the seeds of each accession were sown in 50 cavity trays previously prepared with Peat moss as a substrate. Following sowing, a fungicide suspension (3.0 mL per cavity of Intercaptan^®^ 2.0 g/L) was applied. The seedlings were watered, covered with black plastic, and incubated in darkness at room temperature until they emerged. Once the seedlings emerged, the plastic was removed, and the trays were maintained under a photoperiod of 16 h light/8 h darkness at a temperature of 28 ± 2 °C until the plants developed 4 to 6 true leaves. Virus inoculation was then performed using bioballistics [20].

### 4.3. Viral Clones and Inoculum Preparation

The viral clones of PepGMV and PHYVV used for inoculation were kindly donated by Dr. Rafael Rivera Bustamante, from the Laboratorio de Virología Vegetal of the CINVESTAV-IPN Irapuato. These correspond to infectious dimers of PepGMV and PHYVV, constructed and described by Carrillo-Tripp et al. [20]. For particle preparation, 60 mg of 0.7 μm tungsten particles (Bio-Rad, Hercules, CA, USA) were used. The particles were washed with 2.0 mL of HNO_3_, sonicated for 20 min with H_2_O and ice in Corex glass tubes, and then transferred to 1.5 mL microcentrifuge tubes. After centrifugation, the HNO_3_ was decanted, and 1.0 mL of absolute ethanol was added. The mixture was sonicated and stirred for 1 min, followed by centrifugation at 12,000 rpm for 15 min. Ethanol was discarded, and 1 mL of sterile distilled water was added. Then, 250 μL of the particle suspension was taken, and 700 μL of sterile distilled water was added. To coat the particles with viral DNA (infectious dimeric clones), the suspension was sonicated, and 50 μL was transferred to a new 1 mL tube (sufficient for six bombardments). Then, 5 μL (1 μg/μL) of viral DNA, 50 μL of 2.5 M CaCl_2_, and 20 μL of 0.1 M spermidine were added. The mixture was vortexed for 1 min, sonicated, and centrifuged at 10,000 rpm for 15 min. The supernatant was discarded, and the pellet containing the DNA-coated particles was washed once with 500 μL of absolute ethanol, then sonicated and centrifuged again at 12,000 rpm for 15 min. Finally, the supernatant was discarded, and the pellet was resuspended in 60 μL of ethanol for bioballistic bombardment [2].

### 4.4. Inoculation of Plants with PepGMV and PHYVV by Bioballistics

The infectious clones PepGMV and PHYVV were inoculated separately. The plants of each accession were arranged on individual trays, each corresponding to a specific accession. The number of plants per tray ranged from a minimum of 15 and a maximum of 45 plants, depending on seed availability and germination rate, which among accessions. Each tray was divided into three sections: one for inoculation with PepGMV, another for PHYVV, and a third section with five control plants that were bombarded only with particles lacking viral DNA. This design allowed the response of each accession to infection by both begomoviruses to be evaluated independently.

The particles previously coated with the PepGMV and PHYVV clones were applied using bioballistics device. Inoculation was performed on the first and second youngest leaves; each shot was directed at the center of the leaf blade, near the main vein. The tungsten microparticles were accelerated by helium pressure to 100 or 120 psi using an adapted delivery device [2].

### 4.5. Evaluation of PepGMV and PHYVV Symptoms in Pepper Accessions

The inoculated plants were kept in a growth chamber with a photoperiod of 16:8 (light–dark) at an average temperature of 27 ± 2 °C [16]. Symptom severity caused by the viruses was evaluated qualitatively using the scales proposed by Anaya-López [34] for PepGMV and by Torres-Pacheco [35] for PHYVV, with some modifications (Table 5). Weekly severity ratings were recorded for each plant and accession, starting at 7 days and continuing up to 35 days postinoculation (dpi). Disease progression was documented through photographs taken at 7, 14, 21, 28, and 35 dpi to identify virus-induced symptoms and phenotypic traits, and to relate them to the severity scale.

Disease progression was assessed by calculating the Area Under the Disease Progress Curve (AUDPC), using mean severity values recorded at times 7, 14, 21, 28, and 35 dpi. Based on these values and following the classification proposed by Reyes-Tena et al. [39] with modifications, accessions were categorized into five groups: Resistant, Tolerant, Moderately Susceptible, Susceptible, and Highly Susceptible (Table 6). This classification integrated the slope of the curve and the mean severity at 35 dpi for each accession.

### 4.6. Analysis of Viral Infection by PCR in Asymptomatic Plants

To confirm the efficiency and consistency of the bioballistic inoculation method, molecular detection of viral DNA specific to PepGMV and PHYVV was performed by PCR targeting the Rep and CP genes. A constitutively expressed plant gene (actin) was amplified as an internal control to verify DNA extraction quality and PCR performance.

Tissue samples were collected from distal leaves (new leaves) at 14 dpi from plants of each accession that did not exhibit characteristic viral symptoms. Samples were immediately frozen at −80 °C and pulverized using liquid nitrogen. For DNA extraction, 0.1 g of pulverized tissue was used following the CTAB protocol with modifications [40]. To detect PepGMV and PHYV viral DNA, 20 ng of DNA was used for PCR. Amplification was performed in a thermal cycler SelectCycler™ II (Select BioProducts, Edison, NJ, USA) using virus-specific primers. For PepGMV, a 121 bp fragment of the Rep protein gene was amplified using the forward primer 5′-CAAAGCTGGTGATCCGAAAACG-3′ and the reverse primer 5′-GTTAAACGAGGATAATGGATAAGG-3′. A 104 bp fragment of the CP protein gene was amplified using the forward primer 5′-CCCATCGTGTAGGCAAGCGTTTCTG-3′ and the reverse primer 5′-CATGACCTGTGTGTGTGGTGTGTGTGTGTTCTTG-3′ [20]. For PHYVV, a 142 bp fragment of the Rep protein gene was amplified using the forward primer 5′-CGTCTCCCTCAACTACAAAACC-3′ and the reverse primer 5′-ATCGGTTGTTCGTGCATTGG-3′. For the CP protein gene, a 99 bp fragment was amplified using the forward primer 5′-CCTCAGCTTGGGTTAATCGC-3′ and the reverse primer 5′-CCTTACAGGGACCTTCACAACC-3′ [41]. PCR conditions for both viruses were as follows: an initial denaturation at 94 °C for 5 min, followed by 35 cycles of 94 °C for 30 s, 58 °C for 30 s, and 72 °C for 30 s, with a final extension at 72 °C for 10 min. The PCR products were separated by electrophoresis on 2% agarose gels, stained with ethidium bromide, and visualized under ultraviolet light using a DyNA Dual-Intensity UV Transilluminator.

### 4.7. Statistical Analysis

The weekly data on symptom severity, recorded by plant and accession, were organized into a database for statistical analysis. The mean weekly severity per accession caused by each virus was estimated, along with the percentage of incidence (% of plants with severity > 0). Based on the data on mean severity and incidence, the Area Under the Disease Progression Curve (AUDPC) was calculated using the formula proposed by Pedroza and Samaniego [41]. AUDPC values were analyzed using the Kruskal–Wallis test to detect significant differences in disease progression and total disease accumulation among accessions.

The response of pepper accessions to viral inoculation was analyzed independently for each virus (PepGMV and PHYVV). For each case, a two-way factorial ANOVA was performed considering the effects of accession and time (dpi) on symptom severity. Differences among accessions at 35 dpi were analyzed using Tukey’s test (*p* < 0.05). The effect of domestication level on virus-induced severity was assessed by grouping accessions into domesticated, semidomesticated, and wild accession categories. Differences in severity at 35 dpi among these groups were analyzed using one-way ANOVA followed by Tukey’s test (*p* < 0.05).

All statistical analyses were conducted using R software (version 4.3.3), and graphs were generated with GraphPad Prism (version 10.5.0).

## 5. Conclusions

This study revealed significant variability in symptom severity, incidence, and area under the disease progression curve (AUDPC) among *C. annuum* accessions inoculated with the begomoviruses PepGMV and PHYVV. Wild and semidomesticated accessions exhibited lower susceptibility compared to domesticated ones, suggesting that the degree of domestication influences resistance levels to these viral pathogens. However, differences were not always statistically significant across all groups, particularly for PepGMV, which may be attributed to phenotypic variability within domestication groups and unequal sample sizes. Notably, the wild accession Acc-106 (from Tabasco) showed resistance to PepGMV and tolerant behavior against PHYVV, highlighting its potential as a valuable genetic resource for breeding programs focused on viral resistance. Overall, this work underscores the importance of Mexican wild germplasm as a potential source of resistance genes against begomoviruses in pepper.

## Figures and Tables

**Figure 1 plants-14-02708-f001:**
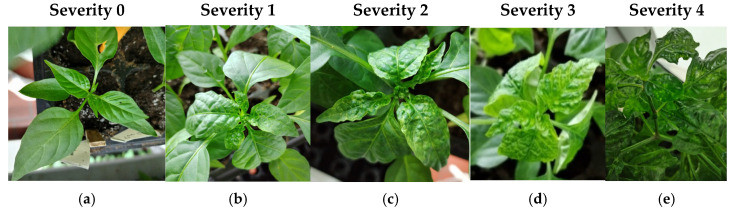
Pictorial scale of symptom severity in pepper plants inoculated with PepGMV. (**a**) Severity 0: asymptomatic plants; (**b**) Severity 1: initial symptoms; (**c**) Severity 2: moderate symptoms; (**d**) Severity 3: severe symptoms; (**e**) Severity 4: very severe symptoms. Photo credit: own elaboration.

**Figure 2 plants-14-02708-f002:**
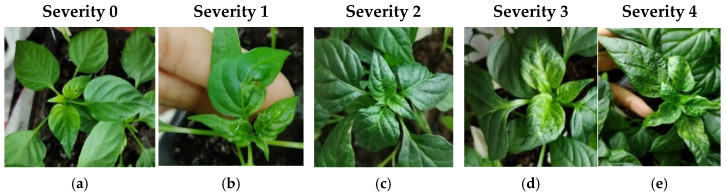
Pictorial scale of symptom severity in pepper plants infected with PHYVV: (**a**) Severity 0: asymptomatic plants; (**b**) Severity 1: initial symptoms; (**c**) Severity 2: moderate symptoms; (**d**) Severity 3: severe symptoms; (**e**) Severity 4: very severe symptoms. Photo credit: own elaboration.

**Figure 3 plants-14-02708-f003:**
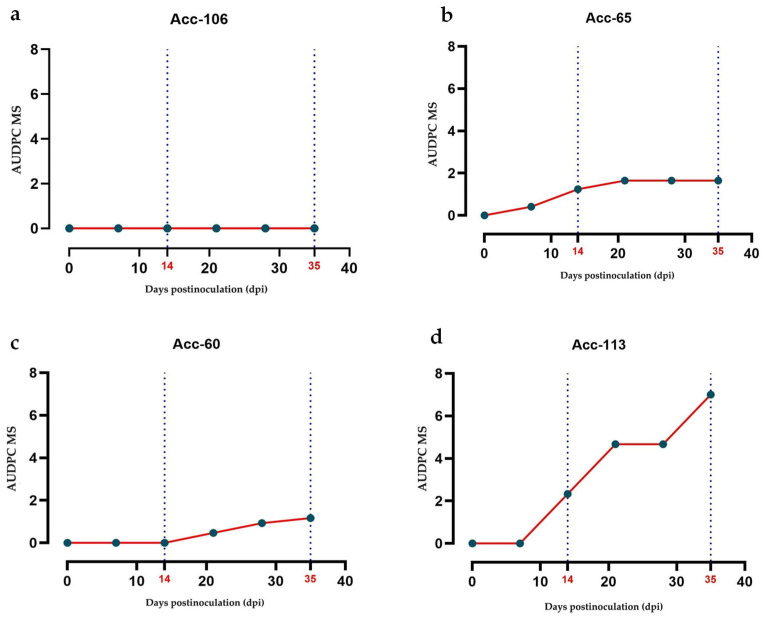
Dynamics of the area under the disease progression curve (AUDPC) for mean severity caused by PepGMV in four *Capsicum annuum* accessions with different susceptibility levels: (**a**) Acc-106 (wild) Resistant; (**b**) Acc-65 (domesticated) Tolerant; (**c**) Acc-60 (domesticated) Tolerant; (**d**) Acc-113 (domesticated) Moderately susceptible.

**Figure 4 plants-14-02708-f004:**
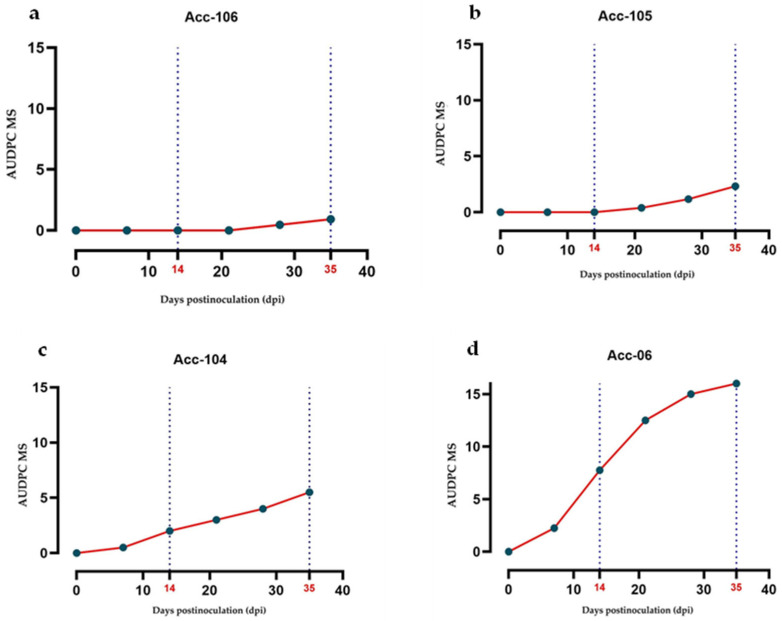
Dynamics of the area under the disease progression curve (AUDPC) for mean severity caused by PHYVV in four *Capsicum annuum* accessions with different susceptibility levels: (**a**) Acc-106 (wild) Tolerant; (**b**) Acc-105 (wild) Tolerant; (**c**) Acc-104 (wild), Moderately Susceptible; (**d**) Acc-06 (domesticated) Susceptible.

**Figure 5 plants-14-02708-f005:**
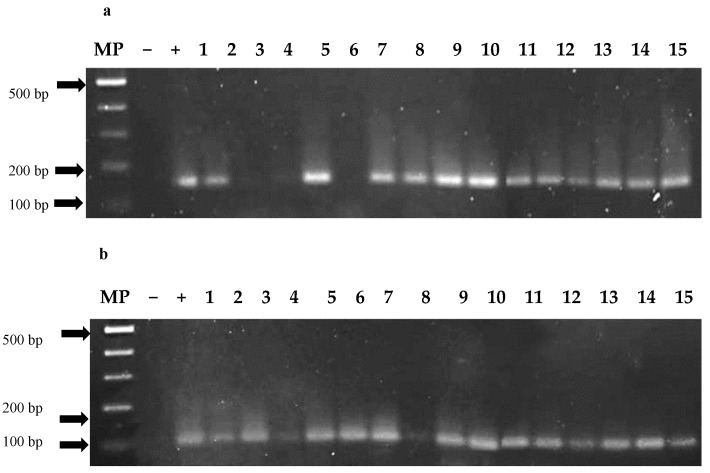
PCR identification of PepGMV viral DNA in 15 asymptomatic Acc-106 pepper plants. (**a**) Amplification of the Rep protein fragment (121 bp) and (**b**) Amplification of the CP protein fragment (104 bp). Samples taken 14 days postinoculation. Lane 1: 100 bp marker, Lane 2: Negative control (bombarded sheets with no viral DNA). Lane 3: Positive control (symptomatic leaves). Lanes with numbers 1 to 15 correspond to the asymptomatic plants of Acc-106.

**Table 1 plants-14-02708-t001:** Influence of the level of domestication on the response to PepGMV and PHYVV in pepper accessions at 35 days postinoculation.

Level of Domestication.	*n*	PepGMV			PHYVV		
		MS ^i^	Min–Max ^iii^	% Inci.	MS	Min–Max	% Inci.
Domesticated	10	0.66 ± 0.07 a ^ii^	0.20–1.33	45	1.30 ± 0.07 b	0.88–2.30	86
Semi-domesticated	1	0.50 ± 0.27 a	0.50	30	0.90 ± 0.26 ab	0.90	80
Wild	4	0.44 ± 0.13 a	0–1.11	21	0.61 ± 0.12 a	1.15–0.13	34

*n* = total accessions that make up the domestication group. ^i^ Mean severity ± standard error. ^ii^ Means with the same letter within the same column are not significantly different according to Tukey’s test (*p* ≤ 0.05). ^iii^ Minimum and maximum value of mean severity between accessions that make up each domestication group.

**Table 2 plants-14-02708-t002:** Mean severity and incidence (%) of symptoms at 35 days postinoculation in 15 accessions of *Capsicum annuum* inoculated with *Pepper golden mosaic virus* (PepGMV) and *Pepper huasteco yellow vein virus* (PHYVV).

Accession	Type or Breed	DD	PepGMV				PHYVV			
			TSP/TPI	Rank Min-Max	MS ^i^ (0–4)	% Inci.	TSP/ TPI	Rank Min-Max	MS (0–4)	% Inci.
Acc-06	Guajillo	D	7/15	0–2.0	0.87 ± 0.19 ab ^ii^	47	13/14	0–3.0	2.30 ± 0.20 a	93
Acc-12	Puya	D	12/18	0–2.0	0.89 ± 0.18 ab	67	17/19	0–2.0	1.16 ± 0.17 b	89
Acc-21	Puya	D	9/17	0–1.0	0.52 ± 0.18 ab	53	14/17	0–2.0	1.06 ± 0.18 bc	82
Acc-50	Ancho	D	6/12	0–2.0	0.92 ± 0.22 ab	50	10/11	0–2.0	1.09 ± 0.23 bc	91
Acc-60	Huacle	D	3/15	0–1.0	0.20 ± 0.20 ab	20	13/15	0–3.0	1.33 ± 0.20 ab	87
Acc-62	Huacle	D	6/16	0–2.0	0.62 ± 0.19 ab	38	12/16	0–4.0	1.31 ± 0.19 b	75
Acc-63	Serrano CM334	SD	3/10	0–2.0	0.50 ± 0.24 ab	30	8/10	0–2.0	0.90 ± 0.24 bc	80
Acc-65	Cascabel	D	3/17	0–2.0	0.24 ± 0.19 ab	18	17/18	0–2.0	1.06 ± 0.18 bc	94
Acc-66	Cora	D	10/16	0–2.0	0.81 ± 0.20 ab	63	15/16	0–2.0	1.38 ± 0.19 ab	94
Acc-67	Chile de agua	D	3/8	0–2.0	0.75 ± 0.27 ab	38	4/8	0–2.0	0.88 ± 0.27 bc	50
Acc-70	Chiltepín	W	3/12	0–2.0	0.33 ± 0.22 ab	25	9/13	0–2.0	1.15 ± 0.21 b	69
Acc-104	Amashito	W	2/7	0–4.0	0.71 ± 0.30 ab	29	2/7	0-3.0	0.85 ± 0.29 bc	29
Acc-105	Amashito	W	4/9	0–3.0	1.11 ± 1.26 a	44	2/9	0-2.0	0.44 ± 0.25 bc	22
Acc-106	Pico Paloma	W	0/15	0	0 b ± 0.20 b	0	2/15	0–1.0	0.13 ± 0.20 c	13
Acc-113	Bola	D	4/6	0–2.0	1.33 ± 0.31 a	67	5/5	0-2.0	1.60 ± 0.34 ab	100
Overall average			75/193	0–4.0	0.65 ± 0.10	40	143/193	0–4.0	1.10 ± 0.13	77

DD = Degree of domestication. D = Domesticated, SD = Semidomesticated, and W = Wild. TSP = Total symptomatic plants. TPI = Total of plants inoculated. MS = Mean of the severity calculated with a Tukey test (*p* ≤ 0.05). Inci. = Incidence. ^i^ Mean severity ± standard error. ^ii^ Means with the same letter within the same column are not significantly different according to Tukey’s test (*p* ≤ 0.05).

**Table 3 plants-14-02708-t003:** Area under the disease progression curve estimated for mean severity and incidence, and classification of 15 *Capsicum annuum* accessions.

Accession		PepGMV			PHYVV		
	DD	AUDPC MS	AUDPC % Inci.	Classification	AUDPC MS	AUDPC % Inci.	Classification
Acc-06	D	5.83 b ^a^	327	T	16.00 a	650	S
Acc-12	D	6.03 b	447	T	8.11 b	626	MS
Acc-21	D	3.29 c	329	T	7.00 c	556	T
Acc-50	D	5.83 b	350	T	7.32 c	636	T
Acc-60	D	1.17 e	117	T	8.87 b	607	MS
Acc-62	D	3.94 c	241	T	8.53 b	525	MS
Acc-63	SM	2.45 d	175	T	6.30 d	560	T
Acc-65	D	1.65 e	124	T	7.19 c	642	T
Acc-66	D	5.69 b	438	T	9.19 b	656	MS
Acc-67	D	4.81 c	263	T	6.13 d	350	T
Acc-70	W	2.33 d	175	T	8.08 b	485	MS
Acc-104	W	4.00 d	200	MS	5.50 e	200	MS
Acc-105	W	6.61 b	311	MS	2.33 f	156	T
Acc-106	W	0.00 f	0	R	0.93 g	93	T
Acc-113	D	7.00 a	467	MS	7.7 c	700	MS

^a^ Values with the same letter in the same column do not differ significantly according to Dunn’s test (*p* ≤ 0.05). DD: Degree of domestication; D = Domesticated, SD = Semidomesticated, and W = Wild. AUDPC MS: Area under the disease progression curve estimated from mean severity values. AUDPC % Inci.: area under the disease progression curve estimated from the percentage of incidence. Classification: category assigned based on the value of mean severity and behavior of the AUDPC: R = Resistant, T = Tolerant, MS = Moderately susceptible, S = Susceptible, HS = Highly susceptible.

**Table 4 plants-14-02708-t004:** Pepper accessions (*Capsicum annuum*) selected for the study of Begomovirus transmission: PepGMV and PHYVV by Bioballistics.

Accession	Region of Origin	Type or Breed	DD	Handling Category
Acc-6	Zacatecas, Zac.	Guajillo	D	Cultivated Creole
Acc-12	El Saladillo, Zac.	Puya	D	Cultivated Creole
ACC-21	El Saladillo, Zac.	Puya	D	Cultivated Creole
Acc-50	El Saladillo, Zac.	Ancho	D	Cultivated Creole
Acc-60	Oaxaca	Huacle	D	Cultivated Creole
Acc-62	Oaxaca	Huacle	D	Cultivated Creole
Acc-63	Morelos	Serrano CM334	SD	Uncultivated Creole *
Acc-65	Guadalajara, Jal.	Cascabel	D	Cultivated Creole
Acc-66	Nayarit	Cora	D	Cultivated Creole
Acc-67	Oaxaca	Chile de agua	D	Cultivated Creole
Acc-70	Queretaro	Chiltepín	W	Wild
Acc-104	Tabasco	Amashito	W	Wild
Acc-105	Tabasco	Amashito	W	Wild
Acc-106	Tabasco	Pico Paloma	W	Wild
Acc-113	Improved variety	Bola	D	Variety grown

DD = Degree of domestication. D = Domesticated, SD = Semidomesticated, and W = Wild. * Source of resistance to *P. capsici*.

**Table 5 plants-14-02708-t005:** Modified symptom severity scale for *Pepper golden mosaic virus* (PepGMV) and *Pepper huasteco yellow vein virus* (PHYVV).

Severity	Symptoms	
	PepGMV	PHYVV
0	No symptoms	No symptoms
1	Slight distortion on apical leaves and visible yellow dots on leaves. Yellow dots visible in isolated patches on apical leaves.	Presence of dots in isolated groups slightly yellow. Slight wrinkling of the apical leaves.
2	Isolated and yellow mosaics forming a network at the base of the apical leaves.	The groups of isolated points begin to be observed as a network at the base of the apical sheets.
3	Leaves curly in the middle part. Slight curvature of the leaves and slightly distorted wrinkles.	Formation of insula-like bumps in the middle parts of the leaves and begin to curve slightly.
4	Complete distortion of the blades and they are smaller in size with respect to the control.	The curved leaves begin to distort. The leaves of the affected plants are smaller.

Own elaboration

**Table 6 plants-14-02708-t006:** Classification of *Capsicum annuum* germplasm susceptibility based on mean severity and area under the disease progression curve (AUDPC) for *Pepper golden mosaic virus* (PepGMV) and *Pepper yellow vein huasteco virus* (PHYVV) inoculated by Bioballistics.

Severity Level	Behavior of the Area Under the Disease Progression Curve	Classification
0	No slope, no presence of the disease.	Resistan (R)
0.1–1.0	With a slight increase and reaching the asymptote at low levels and then remaining constant.	Tolerant (T)
1.1–2.0	The slope increases over time.	Moderately Susceptible (MS)
2.1–3.0	Steep slope with increments over time.	Susceptible (S)
3.1–4.0	Steep and steep slope from the onset of the disease.	Highly Susceptible (HS)

## Data Availability

All data generated or analyzed during this study are included in this published article, and the other specific data are available upon request from the corresponding author.

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
