# Peer review of "Resistance Assessment to PHYVV and PepGMV in Wild and Domesticated Accessions of Capsicum annuum L. by Bioballistic Inoculation"

_plants, 2025, doi:10.3390/plants14172708_

Round 1

Reviewer 1 Report

Comments and Suggestions for Authors

The manuscript by De Lira-Ramos et al. describes the resistance assessment of different accession of Capsicum annum by a well-done experimental approach. The manuscript fulfils the requirements to be published, however, there are some minor aspects that need to be verified:

79-84 It should also be explained that mechanical inoculation for these Begomoviruses is not easily applicable or not possible.

202-215 In table 2 for PepGMV there is a trend in the Mean severity level that seems to show an effect on the increasing of susceptibility linked to the level of domestication, however the three groups (D, s-D and W) have the same letter so they are not significantly different according with your analysis. May you explain better? Same for lines 373-374, 403 in discussion and 569 in conclusion.

In the results section there are some consideration that should be rather be placed in discussion. See lines 153-156, lines 215-218 and lines 351-354.

454-455 The last sentence seems incomplete.

529-530 Are you shore that DNA extraction was performed by 0,01 grams? Usually, the amount of the sample for DNA extraction by CTAB is 0,1 grams. Did you start with dried tissues material?

574-575 The sentence is not clear, please rephrase.

Reviewer 2 Report

Comments and Suggestions for Authors

First of all, I would like to say that I think the study was a success. It was easy to read, but I do have a few suggestions or requests for changes that I think are important. In my view, an alternative structure for the presentation of results would be useful in order to make the infection efficiency of the biolistic bombardment more comprehensible. I would suggest presenting the results first on a high-susceptibility reference line. This acts as an internal control and provides an important basis for evaluating subsequent data from different accessions. In this way, it could be shown early on in the results section that the chosen inoculation method in principle reliably leads to virus infection. Because, only at the end of the manuscript, it is written, that the authors also examined asymptomatic plants using PCR and were thus able to detect a real infection rate of over 90 % - a strong indication of the high efficiency of the biolistic approach. The tested sample size of 15 plants per accession appears fundamentally appropriate in view of this high rate. Because my first question was, are 15 plant per accession sufficient, if infection rate is low or moderate?! Apparently, it is quite robust or high, therefore my concern about statistics is satisfied. However, I still see potential for improvement in the molecular analysis. Although a qualitative PCR was used, it lacked an internal plant amplification control. Regardless of whether a qPCR or endpoint PCR is performed, it would have made methodological sense to amplify a housekeeping gene (e.g. actin or EF1α or any plant gene) as a reference. Ideally, this could have been realized as part of a dual or even triplex PCR - together with virus-specific target regions such as Rep and the coat protein. Such a strategy ensures that negative PCR results are actually due to the absence of viral nucleic acids and not to technical problems during extraction or amplification. Nevertheless, I strongly recommend to perform qPCR analysis in the future.

The conclusion says “Notably, the wild  accession Acc-106 (from Tabasco) showed resistance to PepGMV and tolerant behavior  against PHYVV, highlighting its potential as a valuable genetic resource for breeding…”, therefore I assume that, non of these were infected or that no PCR was positive, respectively. So I conclude, that the study reports that this certain accession was classified as "resistant" because no symptoms occurred and no viral replication was detectable. However, although a sample of 15 plants was also examined here, which is basically acceptable as mentioned above, I would have liked to have seen more in-depth validation for this particularly interesting accession. This must be done by repeating the experiment with a larger sample set, ideally supplemented by an alternative inoculation method. Only then could the postulated resistance be proven with greater certainty.

So, to be clear again, I think it is necessary to visualize the tables with graphs where it makes sense. I would start the manuscript with a typical course of an infection with a susceptible reference line. In addition, graphs showing the severity at certain points in time, plus the images (which accession is shown in Figures 1 and 2?), with statistics, plus qPCR progression. Such a baseline analysis is mandatory at the beginning and then you can also show that the inoculation method is solid and thus statistics are correct. Please describe exactly which tests were used and why in the material and methods.

Round 2

Reviewer 2 Report

Comments and Suggestions for Authors

Good job. Thank you!